# Porcine Epidemic Diarrhea Virus Infection Induces Autophagosome Formation but Inhibits Autolysosome Formation during Replication

**DOI:** 10.3390/v14051050

**Published:** 2022-05-15

**Authors:** Jae-Yeon Park, Jihoon Ryu, Eui-Ju Hong, Hyun-Jin Shin

**Affiliations:** 1Laboratory of Infectious Diseases, College of Veterinary Medicine, Chungnam National University, Daejeon 13434, Korea; wodus5818@cnu.ac.kr (J.-Y.P.); ejhong@cnu.ac.kr (E.-J.H.); 2Research Institute of Veterinary Medicine, Chungnam National University, Daejeon 13434, Korea; jihoon0511@cnu.ac.kr

**Keywords:** autophagy, autophagy flux, PEDV replication

## Abstract

In this study, we investigated the correlation between the mechanism involved in porcine epidemic diarrhea virus (PEDV) replication and autophagic flux. In this study, we found that as PEDV replicated, production of LC3-II was significantly induced up to 24 h post-infection (hpi). Interestingly, although there was significant production of LC3-II, greater p62 accumulation was simultaneously found. Pretreatment with rapamycin significantly induced PEDV replication, but autolysosome formation was reduced. These results were confirmed by the evaluation of ATG5/ATG12 and LAMP1/LAMP2. Taken together, we conclude that PEDV infection induces autophagosome formation but inhibits autolysosome formation during replication.

## 1. Introduction

Porcine epidemic diarrhea virus (PEDV) infects pigs of all ages and causes enteric diseases such as acute watery diarrhea, vomiting, and dehydration [1]. In particular, infected neonatal piglets show high mortality [2]. Since the first identification of the classic strain (G1) in 1978, the annual outbreak of PEDV infections has been consistently reported worldwide [3]. In Korea, a massive epidemic between 2013 and 2014 identified a strong pathogenetic strain (G2) that was closely related to strains from China and the US [4,5].

PEDV is a positive single-stranded RNA virus with an approximately 28 kb genome containing a 5′ cap and a 3′ poly (a) tail [6]. The genome comprises at least seven open reading frames (ORFs) that encode 16 nonstructural proteins (NSPs) and four structural proteins: spike (S), envelope (E), membrane (M), nucleocapsid (N), and accessory (ORF3) [7].

Autophagy is a conserved intracellular degradation process in which damaged cellular organelles, long-lived proteins, and invading microbes such as viruses are delivered to lysosomes for degradation [8]. Autophagy can be induced by various intracellular and extracellular stimuli such as starvation, hypoxia, endoplasmic reticulum (ER) stress, pathogen-associated molecular patterns (PAMPs), and pathogen infections [9]. In the past decade, a number of studies have suggested that autophagy is closely related to viral life cycles [10]. Many studies have shown that autophagy is an important host defense mechanism that negatively affects viral replication and eliminates virions by lysosomal degradation [11]. Some viruses have mechanisms to block the autophagic machinery or even control it for their effective replication, such as the herpes simplex virus-1 (HSV-1) ICP34.5 and the Us11 protein, which are reported to be autophagy inhibitors. The human immunodeficiency virus 1 (HIV-1) Gag, Env, Nef, and Tat proteins control various autophagic pathways and their effective replication [12,13,14]. Furthermore, autophagy is believed to play a role in viral replication platforms, such as double-membrane vesicles (DMVs), and it similarly utilizes the membranes of autophagosome-like vesicles for viral replication [15,16]. Coronaviruses (CoVs) have long been known to interact with the cellular macroautophagy pathway to promote their replication [17]. They rely on the formation of replication complexes such as DMVs, where viral replication and transcription occur. Therefore, CoV replication is strongly related to the autophagy pathway [18].

Autophagic flux is a measure of autophagic degradation activity and refers to the entire process of autophagy, including autophagosome formation, maturation, and fusion with lysosomes [19]. Autophagosomes are double-membrane structures composed of LC3 (microtubule-associated protein 1 light chain 3) and are widely used as marker proteins [20]. The conversion of LC3-I to LC-3II is a good marker to monitor the occurrence of autophagosome formation [21]. P62/SQSTM1 functions as a receptor for cargo proteins located in autophagosomes and is ultimately degraded once autolysosomes are successfully formed [22,23]. The fusion between autophagosomes and lysosomes induces autolysosome formation; in this case, the p62 protein degrades together with the cargo protein [23]. Therefore, the accumulation of p62 has been used as a marker for the inhibition of autophagy [24]. Lysosome-associated membrane glycoprotein (LAMP) is an important regulator of the autophagy pathway that is delivered to phagosomes during the formation of autolysosomes [25]. LAMP-1 and LAMP-2 are estimated to contribute to approximately 50% of all proteins in the lysosomal membrane [26].

There have been few studies on the role of autophagy in PEDV replication. For example, Guo et al. reported that PEDV infection induced autophagy and positively impacted replication [27]. Lin et al. also reported that autophagy has a positive effect on replication and found that the PEDV nsp6 protein worked as an autophagy inducer [28]. In contrast, Ko et al. reported that upregulated autophagy negatively affected replication [29]. Although three conflicting reports have been published, they concluded that these differences were attributable to different strains of PEDV.

In this study, we further studied the role of autophagy in PEDV replication and found some results that differed from previous reports.

## 2. Materials and Methods

### 2.1. Cells and Viruses

African green monkey kidney cell lines (Vero) were cultured in Minimum Essential Media (MEM; biowest, Nuaillé, France) supplemented with 10 mM HEPES (Gibco, Carlsbad, CA, USA) and 10% antibiotic-antimycotic (Gibco, Carlsbad, CA, USA) fetal bovine serum (Gibco, Carlsbad, CA, USA) at 37 °C with 5% CO_2_. In this study, we used the PEDV strain PED-CUP-B2014 [30]. Virus titer was determined by TCID_50_ assay [31].

### 2.2. Virus Infection and Titration

For autophagy induction, Vero cells were pretreated with 100 nM of rapamycin (Abcam, Cambridge, MA, USA) for 8 h and then washed 3 times with PBS. After washing, PEDV was infected with multiplicity of infection (MOI) of 1.0. After 1 h incubation at 37 °C, it was washed 3 times with PBS and replaced with fresh MEM medium. After infection, the supernatant was collected, and the PEDV titer was determined by plaque assay. The plaque assay plates were scanned and counted by CTL ELISpot reader (Cellular Technology Limited, Shaker Heights, OH, USA).

### 2.3. Plasmid and Antibodies

The pcDNA EGFP-RFP-LC3B plasmid encoding EGFP, RFP, and LC3B was constructed by the pcDNA 3.1 myc-his (-) vector. The pcDNA EGFP-mRFP-LC3B (ptfLC3) plasmid encoding *Chlorocebus sabaeus* LC3B (NCBI Reference Sequence: XM_007994295.2) was constructed by the pcDNA 3.1 myc-his (-) vector. Anti-LC3B antibody (Cell Signaling, Danvers, MA, USA, #2775), anti-SQSTM1/p62 antibody (Cell Signaling, #5114), anti-ATG5 antibody (Cell Signaling, #9980), anti-ATG12 antibody (Cell signaling, #4180), anti-LAMP1 (Cell Signaling, #3243), anti-β-actin (Santa Cruz, sc-47778), and mouse anti-PEDV antibody were made in our laboratory (immunized inactivated PEDV in mouse).

### 2.4. Western Blot Analysis

The cells were washed with cold PBS and lysed in RIPA buffer (Thermo Fisher Scientific Inc., Waltham, MA, USA) containing protease inhibitor cocktail (Sigma Aldrich, Steinheim, Germany) for 30 min at 4 °C, and then the supernatant was collected. The protein concentration was determined with a BCA protein assay kit (Thermo Fisher Scientific Inc., Waltham, MA, USA). SDS–PAGE and Western blotting were carried out using standard methods. Briefly, equivalent amounts of protein were separated on polyacrylamide-tricine gels (15% polyacrylamide). After SDS–PAGE, the gels were transferred onto 0.45 μm polyvinylidene fluoride (PVDF) membranes (Millipore, Billerica, MA, USA), followed by blocking with 5% BSA in TBST (TBS with 0.1% Tween 20) for 1 h at room temperature. The membrane was incubated with primary antibody at 4 °C overnight. After being washed with TBST, the membrane was incubated with HRP-tagged anti-rabbit IgG (1:10000 dilution) for 2 h at room temperature. The images were observed with ECL solution (SuperSignal West Femto Maximum Sensitivity Substrate 34095) using an ATTO Luminograph (Japan).

### 2.5. Quantitative Real-Time PCR

Vero cells were seeded in 6-well plates. The cells were washed with PBS and lysed in RiboEX Total RNA (GeneAll, Seoul, South Korea), and reverse transcription was performed using BioFACT™ 2X RT Pre-Mix (BioFACT™, Korea) amplified with random hexamer (Thermo Fisher Scientific Inc., Waltham, MA, USA). Amplification was carried out in a 20 μL reaction mixture containing 10 μL TOPreal™ qPCR 2X premix (Enzynomics, Daejeon, South Korea), 0.2 μM concentration of each primer (Table 1), and 1 μL cDNA. The reaction procedure was 95 °C for 5 min, followed by 40 cycles at 95 °C for 30 s, 58 °C for 30 s, and 72 °C for 30 s. The relative mRNA expression level was normalized to the housekeeping gene GAPDH. The relative transcript levels were analyzed using the ΔΔ Ct method.

### 2.6. Confocal Microscopy

We followed the general manual for confocal microscopy. Briefly, Vero cells were seeded in 12-well plates. The cells were treated with PED-CUP-2014B (MOI = 1). At 18 h infection, the cells were fixed with 4% paraformaldehyde for 15 min. Following fixation three times washed in PBS, the cells were then permeabilized with 0.25% Triton X-100 for 10 min and blocked in PBS containing 2% BSA for 1 h at room temperature. After three washes with PBS, the coverslips were incubated with primary antibodies in PBS containing 2% BSA at 4 °C (overnight). After washing with PBS, the nuclei were stained on Hoechst 33258 (Thermo Fisher Scientific Inc., Waltham, MA, USA). After staining for 15 min, the cells were washed with PBS and mounted onto microscope slides. Fluorescence signals were observed under confocal microscopy. Transfected samples were also subjected to the same procedure.

### 2.7. Gene Knockdown by siRNA

The siRNAs purchased from GenePharma (Shanghai, China) were designed to bind with endogenous ATG5 (Genebank accession number XM_008019454.1) and LAMP1 (Genebank accession number XM_007960989.1). Vero cells were seeded into 6-well plates, and the transfection mixture was prepared with 200 μL of Opti-MEM medium (Gibco, Carlsbad, CA, USA) containing lipofectamine 3000 (Invitrogen, Carlsbad, CA, USA) and 100 nM of each siRNA (Table 2). At 48 h post-transfection, the cells were prepared for various experiments. The silencing efficiency was determined by Western blot assay.

### 2.8. Statistical Analysis

Data are presented as the mean ± standard deviation (SD). Statistical significance was calculated using SPSS and GraphPad Prism 8. A *p*-value < 0.05 was considered statistically significant. Asterisks in figures indicate statistical significance (* *p* < 0.05, ** *p* < 0.01, *** *p* < 0.001).

## 3. Results

### 3.1. Autophagosome Accumulate in PEDV-Infected Cells

Although the correlation between PEDV and autophagy has already been studied by three groups, each study reached different conclusions [27,28,29]. It has been suggested that the different results arose from differences in the virulence among PEDV strains. For this reason, we examined autophagy activation using our PED-CUP-B2014 strain. First, we evaluated the conversion from LC3-I to LC3-II in infected cells. The Western blot results confirmed that PEDV infection significantly induced LC3-I conversion, indicating that autophagy was strongly upregulated (Figure 1A–C). Additionally, to understand how PEDV infection induces autophagy, we check autophagosome formation in cells, as observed by immunofluorescence assay. The results indicated that PEDV-infected cells observed more autophagosome (green) formation compared with mock-infected Vero cells (Figure 1D,E). Based on these results, autophagy induction was positively correlated with PED-CUP-B2014 infection.

### 3.2. Induction of Autophagy with Rapamycin Upregulates the Replication of PEDV in Vero Cell

In a further study, we administered *rapamycin,* which is a specific inhibitor of mTOR. As in our preliminary study, 100 nM rapamycin treatment for 8 h showed the best results in Vero cells. As shown in Figure 1D, rapamycin pretreatment induced both production of LC3-II and PEDV protein expression compared with those of the DMSO treatment as the negative control. We also found that after rapamycin treatment, there was a significant accumulation of p62 with PEDV infection compared with the noninfected control (Figure 2A–D). We also performed a plaque assay for viral titer for comparison with rapamycin treatment. We found an approximately two-fold increase in the PEDV titer with rapamycin treatment (Figure 2E,F).

### 3.3. PEDV Infection Suppresses Autophagic Flux in a Time-Dependent Manner

Next, we examined whether PEDV infection induces autophagic flux. As shown in Figure 3, PEDV infection induced autophagic flux in a time-dependent manner until 24 hpi; both autophagosome formation and the accumulation of p62 were found (Figure 3A–D). As PEDV N protein expression was detected at 8 hpi, the conversion to LC-II began. This was a strong, time-dependent correlation and indicated that as conversion to LC-II increased, PEDV N protein expression also increased. We also observed p62 accumulation from 8 hpi to 24 hpi. Based on these results, we confirmed that PEDV infection induced autophagic flux, but autolysosome formation was inhibited. Therefore, we concluded that PEDV infection induced autophagosome formation.

### 3.4. PEDV Infection Induced Autophagosome Formation but Suppresses Its Fusion with Lysosome

We also reconfirmed this correlation using a tandem-reporter construct, EGFP-mRFP-LC3 (ptfLC3), described in the confocal microscopy in the Materials and Methods section. The EGFP fluorescence from the ptfLC3 fusion protein is substantially quenched in acidic autolysosomal conditions; however, the mRFP fluorescence signal produced by ptfLC3 is not sensitive to acidic conditions. When both EGFP and mRFP signals were co-localized, the result were yellow puncta. As shown in Figure 4D, there were very few autophagosome spots in DMSO-treated cells (the negative control). In contrast, there were many strong autophagosome spots in rapamycin-treated cells (the positive control) (Figure 4F,G). As expected, we found almost equal numbers of green and red spots in rapamycin-treated cells, which demonstrated that autophagic flux was normal (Figure 4H). In contrast, PEDV infection strongly induced autophagosome formation and maturation (Figure 4J,K). The combined fluorescent signals of both fluorophores, red and green, are clearly shown together in the same location such that the autophagosomes appear as yellow puncta (Figure 4L). This indicates that although autophagosomes were successfully formed, they failed to form autolysosomes in PEDV-infected cells. Taken together, these results clearly confirm that PEDV infection successfully induced autophagic flux, but the process was not completed because induction of autophagosome formation was successful but autolysosome formation was not successful.

### 3.5. PEDV Infection Regulated Autophagy-Related Gene mRNA

We also evaluated ATG5 and ATG12 mRNA expressions, which are related to the initiation of autophagosome formation. As shown in Figure 5B, ATG5 mRNA expression was highly upregulated from 4 hpi and gradually decreased until 12 hpi. However, the expression suddenly peaked at 24 hpi. As shown in Figure 5C, ATG12 mRNA expression showed a pattern that was very similar to that of ATG5. Expression was upregulated from 4 hpi and gradually decreased until 12 hpi, but it suddenly peaked at 24 hpi. Based on these results, we concluded that PEDV infection strongly upregulated both ATG5 and ATG12 mRNA expression from 4 hpi and peaked at 24 hpi. Interestingly, we found that both genes were upregulated twice by PEDV infection: in the very early (4 hpi) and late stages (24 hpi). These results confirmed that autophagosome formation is important for PEDV infection during the very early stage and the late stage, such as the budding stage.

In the next step, we investigated why autolysosome formation was inhibited in PEDV-infected cells. To answer this question, we examined whether autolysosome fusion was inhibited by PEDV infection. LAMP1 is well-known as an important factor for autolysosome fusion [25]. We evaluated whether PEDV infection inhibited autolysosome fusion by quantifying LAMP1 expression. As shown in Figure 5D, PEDV infection downregulated LAMP1 mRNA expression at 4 hpi but upregulated it at 8 hpi. Interestingly, expression was significantly downregulated again at 12 hpi but significantly upregulated again at 24 hpi. This up and down pattern was repeated every 4 h. These results indicate that PEDV infection downregulated LAMP1 mRNA expression at both 4 and 12 hpi but significantly upregulated it at 8 and 24 hpi. We do not yet know why the LAMP1 mRNA level increased and decreased every 4 h. We speculate that PEDV infection induced autophagosome formation but inhibited autolysosome formation by inhibiting autolysosome fusion. However, because the RNA level was not consistent, we could not confirm autolysosome fusion inhibition. Therefore, we proceeded to examine the protein level.

### 3.6. PEDV Infection Controls ATG5, ATG12, and LAMP1 Proteins

We also confirmed the correlation of PEDV infection with ATG5, ATG12, and LAMP1 using Western blotting. As shown in Figure 6, we found that ATG5 and ATG12 protein expression was upregulated at 24 hpi compared with that in mock-infected cells. Additionally, we found that LAMP1 expression was downregulated both at 12 hpi and 24 hpi compared with those in mock-infected cells (Figure 6A–G). Based on these results, we concluded that PEDV infection not only induced autophagic flux through the upregulation of ATG5 expression but also reduced LAMP1 expression, which inhibited autolysosome formation.

### 3.7. Effect of Knockdown of ATG5 and LAMP1 on PEDV Replication

To reconfirm our findings, we performed ATG5 and LAMP1 knockdowns using siRNAs. In all knockdown experiments, scrambled siRNA was used as a negative control. The results showed that the knockdowns of ATG5 and LAMP1 using siRNAs significantly affected PEDV protein synthesis (Figure 7A). The knockdown of the ATG5 gene showed a strong reduction in LC3 I conversion compared with the negative control. These results clearly confirmed that with reduced autophagosome formation, PEDV infection strongly increased autophagic flux, but PEDV protein synthesis was strongly reduced. The knockdown of the LAMP1 gene did not affect LC3 I conversion but strongly induced PEDV protein synthesis because LAMP1 is related only to lysosomes. Additionally, we confirmed these results with a plaque assay. The results showed that the knockdown of the ATG5 gene negatively affected PEDV replication, but the knockdown of the LAMP1 gene positively affected PEDV replication (Figure 7G,H). Therefore, we concluded that regulation of both autophagosome formation and autolysosome formation was very important to PEDV viral replication that differed from previous reports.

## 4. Discussion

The autophagy pathway has been shown to play an important role in the replication of many viruses [32,33]. It has been reported that the relationship between autophagy and coronavirus replication is related to the formation of double-membrane vesicles (DMVs), which are associated with the autophagy pathway [18]. For example, the formation of double-membrane complexes was found in mouse hepatitis virus (MHV) infection and was essential for their replication [34]. However, infectious bronchitis virus induces autophagy but does not require autophagy for replication, and transmissible gastroenteritis virus (TGEV) induces autophagy that negatively regulates its replication [35,36]. Although the relationship between PEDV and autophagy has already been previously studied, we further studied the detailed mechanism involved. In the case of PEDV, previous reports have shown that PEDV infection induces autophagy in Vero and IPEC-J2 cells [27,28,29]. The conclusions of the three previous reports differed, and it was suggested that the differences were related to the degree of virulence of the different PEDV strains. In our study, the PED-CUP-B2014 strain induced autophagy (Figure 1A), which is similar to the results of Lin. et al.; replication increased with rapamycin treatment, as shown in Figure 2A. Autophagic flux has been proven to be involved in the processes of autophagosome formation, maturation, and fusion with lysosomes, and this flux is important for many viruses [32,33,37]. Through the accumulation of p62, our Western blot results confirmed that PEDV infection induced autophagosome formation but inhibited autolysosome formation (Figure 3A). This was clearly confirmed by confocal microscopy, as almost all LC3 puncta showed a yellow signal (Figure 4L). Guo X. et al. also found that p62 slightly accumulated during the early life cycle of PEDV infection but not in the late stage. In contrast, we confirmed that PEDV induced autophagosome formation and inhibited autolysosome formation at two points, during the early and late stages of infection.

We also confirmed the correlation between PEDV infection and the autophagy pathway by analyzing autophagy-related genes. It is well known that the ATG5 and ATG12 genes are related to the initiation of autophagosome formation [38]. We confirmed that PEDV infection upregulated both ATG5 and ATG12 gene mRNA expressions, as shown in Figure 4. The LAMP1 genes are known to be related to lysosomal membrane proteins associated with autophagosomes fused to autolysosomes [25,26]. We also confirmed that both genes were downregulated by PEDV infection at 4 hpi and 12 hpi but significantly upregulated at 8 hpi and 24 hpi (Figure 5). An increase in mRNA does not promise increase in protein expression; protein expression is better confirmed by Western blot. Therefore, we confirmed endogenous LAMP1 expression level by Western blot by PEDV infection, and we found LAMP1 expression was reduced at both 12 hpi and 24 hpi, as shown in Figure 6. The reason might be reduced protein expression by viral infection at 24 hpi recovered by the aspect of cell defense mechanism. We also confirmed that PEDV infection induced autophagosome formation and maturation but inhibited lysosome fusion (Figure 3 and Figure 4). Lin. et al. reported that the PEDV nsp6 protein induced autophagy through the PI3K/Akt/mTOR signaling pathway, but they did not detect autophagy inhibition [28]. The results from our study and Guo X. et al. clearly confirmed significant p62 accumulation instead of simple accumulation as a general phenomenon seen during the autophagy process (Figure 3). Therefore, we concluded that PEDV protein(s) may block autolysosome formation. In the case of HIV, Gag protein co-localizes with LC3 to promote virion assembly, and the Nef protein blocks the proteolytic stage of autophagy for replication. [14]. Knockdown of ATG5 clearly reduced PEDV protein expression and viral replication (Figure 7). These results clearly confirmed that PEDV replication was closely related to the autophagy pathway, particularly autophagosome formation. Conversely, the knockdown of LAMP1 increased PEDV protein expression and viral replication. These results clearly confirmed that inhibition of autolysosome formation was beneficial to PEDV replication (Figure 7). In general, autolysosome formation has shown a significant negative effect on viral replication [37]. Similarly, our results also clearly confirmed that PEDV inhibited autolysosome formation.

We found that PEDV infection induced autophagosome formation but inhibited lysosome fusion as regulated by endogenous LAMP1 protein (Figure 8). However, we could not specify which PEDV protein(s) regulated the autophagic flux mechanism. More studies are required to answer this question, which would be very important information for the prevention of PEDV replication.

## Figures and Tables

**Figure 1 viruses-14-01050-f001:**
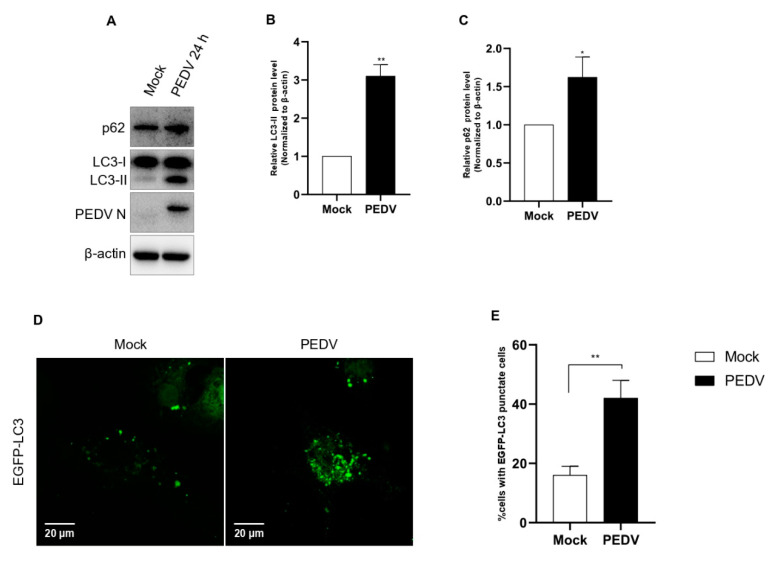
Confirmation of autophagy activation by PEDV infection. (**A**–**C**) Vero cells were infected with PEDV at an MOI of 1 and harvested at 24 hpi. Cell extracts were analyzed with western blotting using anti-LC3B, anti-SQSTM1/p62, anti-PEDV, and anti-β-actin antibodies. β-actin served as an internal control. (**D**) Vero cells were infected with PEDV at an MOI of 1. After 24 hpi, the cells were fixed, and immunofluorescence analysis was performed with anti-LC3B (green) antibody. (**E**) The quantification of cells showing LC3 puncta in PEDV-infected cells. In five random fields, the average number of puncta in each cell. Data are the mean ± SD (*n* = 3; * *p* < 0.05, ** *p*< 0.01). The scale bar indicates 20 μm.

**Figure 2 viruses-14-01050-f002:**
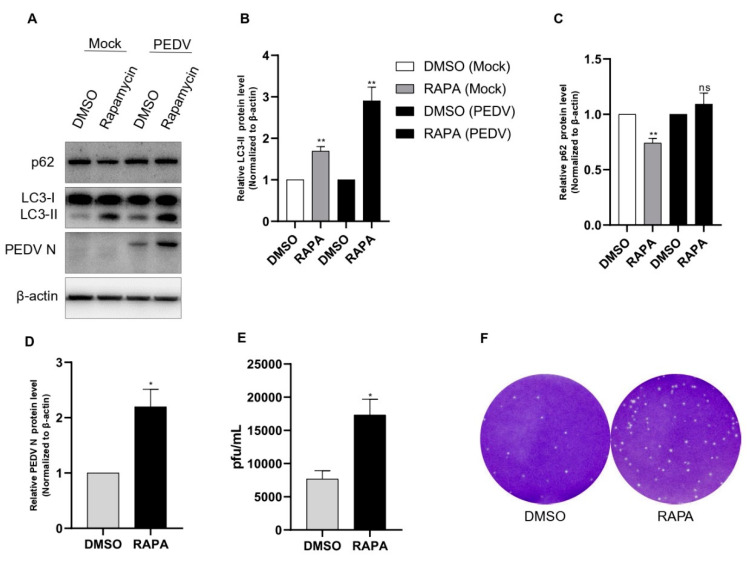
Rapamycin treatment induced PEDV replication. (**A**–**D**) Vero cells were pretreated with rapamycin or DMSO. After 8 h of treatment, the cells were inoculated with PEDV at an MOI of 1. Cell lysates collected at 18 hpi were subjected to a Western blot assay using anti-LC3B, anti-SQSTM1/p62, anti-PEDV, and anti-β-actin antibodies. β-actin served as an internal control. (**E**,**F**) The cells were treated as described in (**A**), and the viral titers of harvested cells were determined at 18 h. Data are the mean ± SD (*n* = 3; * *p* < 0.05, ** *p*< 0.01).

**Figure 3 viruses-14-01050-f003:**
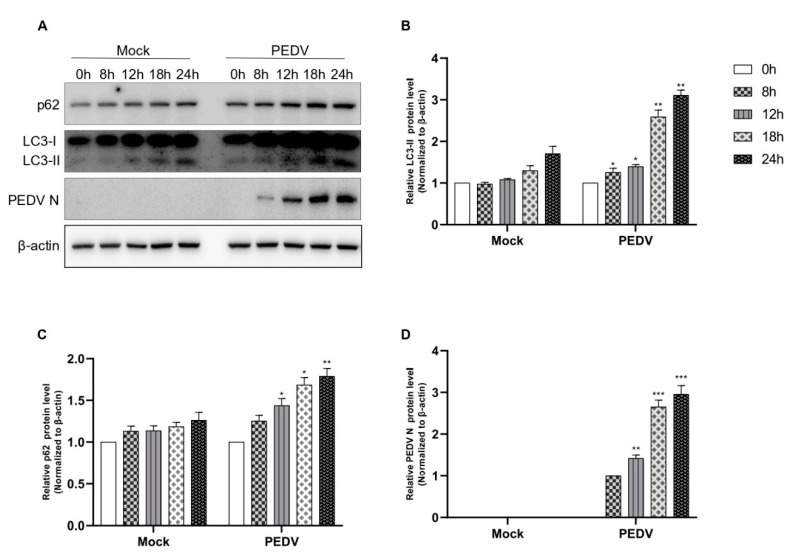
PEDV infection suppresses autophagic flux (**A**–**D**). Vero cells were infected with PEDV at an MOI of 1 and harvested at different times as indicated. Mock represents a negative control. Cell extracts were analyzed with Western blotting using anti-LC3B, anti-SQSTM1/p62, anti-PEDV, and anti-β-actin antibodies. β-actin served as an internal control. Data are the mean ± SD (*n* = 3; * *p* < 0.05, ** *p*< 0.01, *** *p* < 0.001).

**Figure 4 viruses-14-01050-f004:**
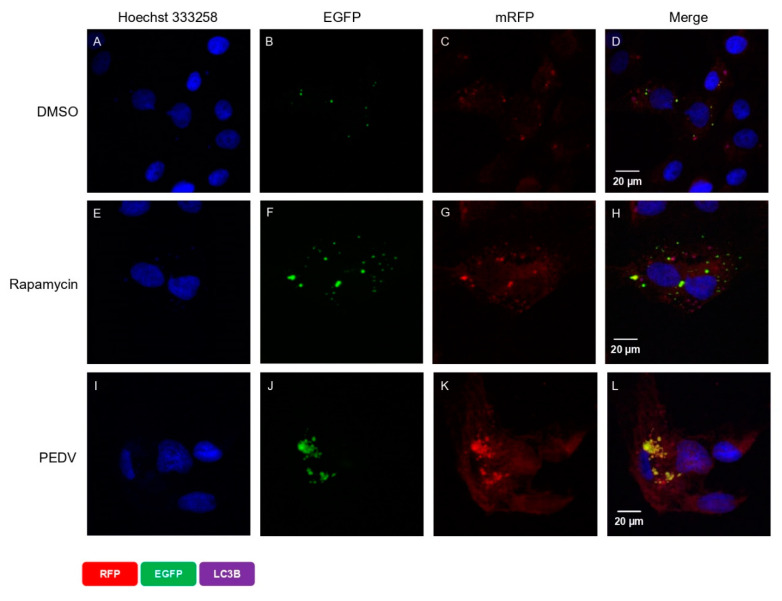
PEDV infection activates autophagosome formation but inhibits its fusion with lysosome. Vero cells were transfected with EGFP-mRFP-LC3B (ptfLC3). After 24 h, the medium was replaced with fresh medium, and the cells were treated with DMSO or rapamycin. After 8 h of treatment, the medium was replaced with new medium, and cells were infected with PEDV at an MOI of 1. After 18 hpi, the cells were fixed, and the nuclei were stained with Hoechst 33258 (blue). (**A**) DMSO-treated control (nucleus), (**B**) DMSO-treated control (EGFP), (**C**) DMSO-treated control (RFP), (**D**) DMSO-treated control (merge), (**E**) rapamycin-treated control (nucleus), (**F**) rapamycin-treated control (EGFP), (**G**) rapamycin-treated control (RFP), (**H**) rapamycin-treated control (merge), (**I**) PEDV-infected cell (nucleus), (**J**) PEDV-infected cell (EGFP), (**K**) PEDV-infected cell (RFP), and (**L**) PEDV-infected cell (merge). The scale bar indicates 20 μm.

**Figure 5 viruses-14-01050-f005:**
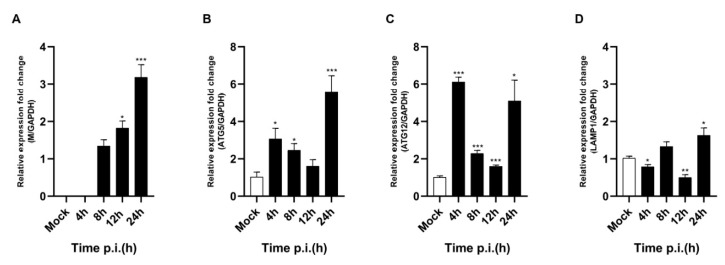
Evaluation of ATG5, ATG12, and LAMP1 mRNA expressions under PEDV infection. (**A**–**D**) Vero cells were infected with PEDV at an MOI of 1 and harvested at different times as indicated. Total RNA was isolated to analyze ATG5, ATG12, LAMP1, and viral M gene mRNA levels using quantitative RT–PCR. The mRNA levels of ATG5, ATG12, and viral M were normalized to the mRNA levels of GAPDH. Data are the mean ± SD (*n* = 3; * *p* < 0.05, ** *p*< 0.01, *** *p* < 0.001).

**Figure 6 viruses-14-01050-f006:**
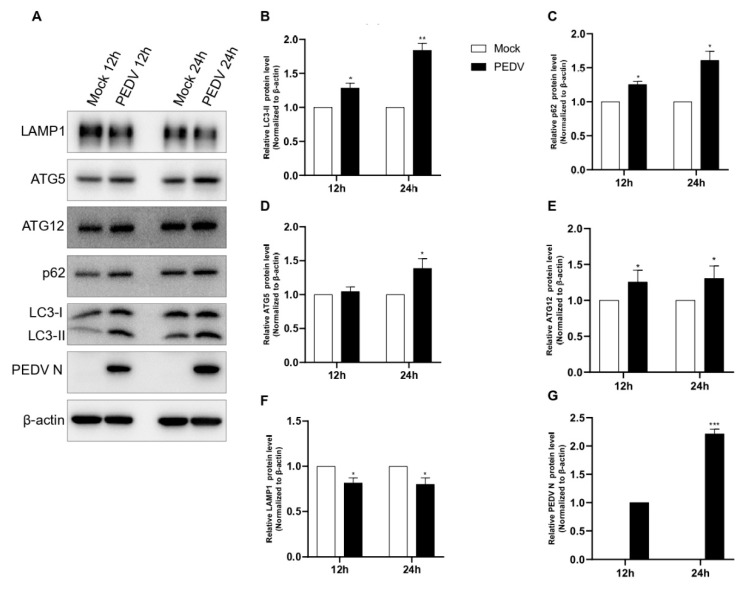
PEDV infection induced ATG5 expression but reduced LAMP1 expression. (**A**–**G**) Vero cells were infected with PEDV at an MOI of 1 and harvested at different times as indicated. Cell extracts were analyzed with a Western blot using anti-LAMP1, anti-ATG5, anti-ATG12, anti-SQSTM1/p62, anti-LC3B, anti-PEDV, and anti-β-actin antibodies. β-actin was used as an internal control. Data are the mean ± SD (*n* = 3; * *p* < 0.05, ** *p*< 0.01, *** *p* < 0.001).

**Figure 7 viruses-14-01050-f007:**
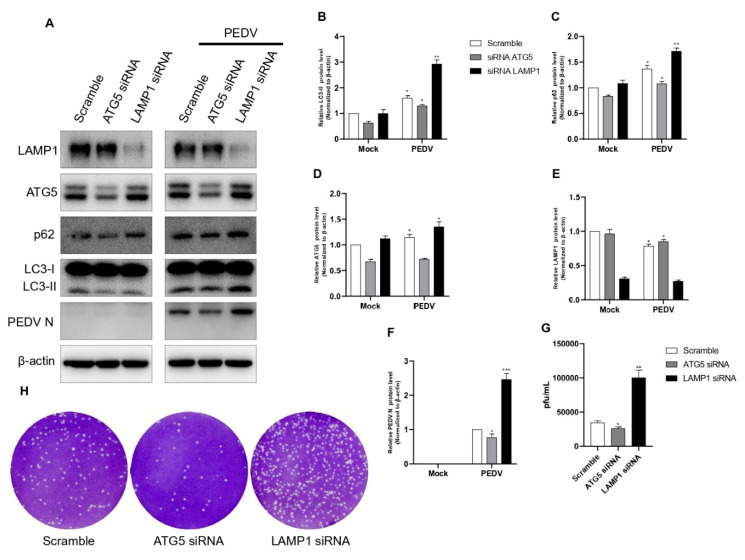
Inhibition of autophagic flux with specific siRNAs targeting the ATG5 and LAMP1 genes reduced PEDV replication. (**A**–**F**) Vero cells were transfected with ATG5, LAMP1, and scrambled siRNA for 48 h. At 48 h post-transfection, cells were infected with PEDV at an MOI of 1 and harvested at 18 hpi. Cell extracts were analyzed with a Western blot using anti-LAMP1, anti-ATG5, anti-SQSTM1/p62, anti-LC3B, anti-PEDV, and anti-β-actin antibodies. β-actin was used as an internal control. (**G**,**H**) The cells were treated as described in (**A**)**,** and the viral titers of harvested cells were determined at 18 h. Data are the mean ± SD (*n* = 3; * *p* < 0.05, ** *p*< 0.01, *** *p* < 0.001).

**Figure 8 viruses-14-01050-f008:**
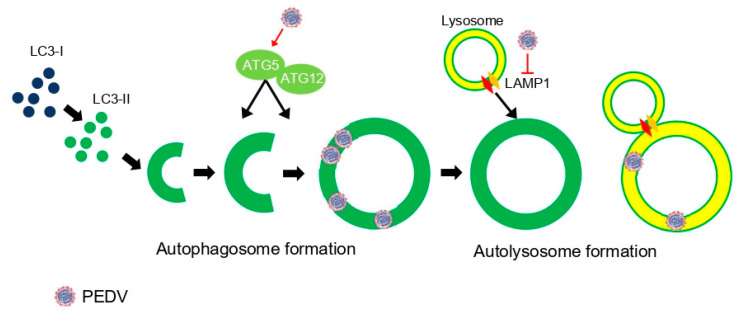
A schematic representation of relationship between PEDV infection and autophagy flux. PEDV infection upregulated autophagy-related genes such as ATG5 and ATG12, which are important for autophagosome formation. However, it inhibited LAMP1, which mediates autolysosome formation in PEDV life cycles.

**Table 1 viruses-14-01050-t001:** qPCR primer sequences.

Primer	Sequence (5′–3′)
qPEDV M F	CGTACAGGTAAGTCAATTAC
qPEDV M R	GATGAAGCATTGACTGAA
qATG5 F	ACCTCTGCAGTGGCTGAGTG
qATG5 R	TCAATCTGTTGGCTGCGGGA
qATG12 F	ACTTGTGGCCTCAGAACAGTTG
qATG12 R	ACCATCACTGCCAAAACACTCA
qLAMP1 F	GTGACCGTAACGCTCCACGA
qLAMP1 R	AGCCTTGTCACGTCGTGTT
qGAPDH F	CCTTCCGTGTCCCCACTGCCAAC
qGAPDH R	GACGCCTGCTTCACCACCTTCT

**Table 2 viruses-14-01050-t002:** siRNA used in this study.

siRNA	Sequence (5′–3′)
ATG5 481 F	GACGUUGGUAACUGACAAATT
ATG5 481 R	UUUGUCAGUUACCAACGUCTT
LAMP1 605 F	CAGGGCAGAUAUAGAUAAATT
LAMP1 605 R	UUUAUCUAUAUCUGCCCUGTT
scramble F	UUCUCCGAACGUGUCACGUTT
scramble R	ACGUGACACGUUCGGAGAATT

## Data Availability

Not applicable.

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
