# Peer review of "Porcine Epidemic Diarrhea Virus Infection Induces Autophagosome Formation but Inhibits Autolysosome Formation during Replication"

_viruses, 2022, doi:10.3390/v14051050_

Round 1

Reviewer 1 Report

Authors investigated the relationship between autophagy and PEDV replication. They claimed that PEDV replication showed a positive correlation with autophagosome formation but a negative correlation with the autolysosome formation. The experimental design is logical and the results are interesting. Minor comments and suggestions to the authors are described below

1 Please describe in detail the statistics method of the quantification of cells showing LC3 puncta in Fig 1 E.

2 Plaque figures should be supplemented in Fig 2 and 7.

3 In Fig 3, LC3-II increased in control, why?

Author Response

Reviewer 1

Authors investigated the relationship between autophagy and PEDV replication. They claimed that PEDV replication showed a positive correlation with autophagosome formation but a negative correlation with the autolysosome formation. The experimental design is logical and the results are interesting. Minor comments and suggestions to the authors are described below

1 Please describe in detail the statistics method of the quantification of cells showing LC3 puncta in Fig 1 E.

-> Thanks for kind comments. We described all the detailed information in materials and method section and also in figure legend.

2 Plaque figures should be supplemented in Fig 2 and 7.

-> Thanks for kind comments. We added information in Figure 2 and 7, also in materials and methods section Line 83 - 89

  1. In Fig 3, LC3-II increased in control, why?

-> Thanks for kind comments. Although we don’t have clear confirmation on this, but we speculate that serum free MEM media one that we used for infection. Because our PEDV strain (PED-CUP-B2014) doesn’t grow well without trypsin and we performed experiments in FBS free condition tried to maximize the trypsin activity. In this environment, we think the cell starvation was induced in mock-infected cells so that LC3-II was activated even in control.

Reviewer 2 Report

Porcine epidemic diarrhea virus (PEDV) causes enteric diseases in pigs. The manuscript submitted by Jae-Yeon Park and co-authors investigates that PEDV infection induces autophagosome formation but inhibits autolysosome formation during replication. Generally, this study is interesting; however, there are still some concerns.

Comments:

  1. In "3.5. PEDV infection regulated autophagy-related gene mRNA", the authors analyzed the mRNA expression levels of autophagy-related genes ATG5, ATG12 and LAMP1. However, the change of mRNA expression level is usually not completely consistent with the change of protein expression level. The protein expression levels of ATG5, ATG12 and LAMP1 should be analyzed by western blot.
  2. Line 212-213, "LAMP1 mRNA expression at 4 hpi but upregulated it at 8 hpi. Interestingly, expression was significantly downregulated again at 12 hpi but significantly upregulated again at 24 hpi"; Line 224-226, "Additionally, we found that LAMP1 expression was downregulated both at 12 hpi and 24 hpi compared with those in mock-infected cells (Figure 6A-F)". Are these two results contradictory? It should be discussed in the discussion or results section.
  3. Electron microscope is the gold standard for judging autophagosome or autolysosome. Electron microscope photos should be supplied in Figure 1, Figure 4 and Figure 7 if possible.
  4. Line 95, "mouse anti-PEDV" should be "mouse anti-PEDV antibody".
  5. Line 143-144, "p value" the "p" should be italic.
  6. Figure 2B and 2C, the Mock and PEDV infection groups should be marked.

Author Response

Porcine epidemic diarrhea virus (PEDV) causes enteric diseases in pigs. The manuscript submitted by Jae-Yeon Park and co-authors investigates that PEDV infection induces autophagosome formation but inhibits autolysosome formation during replication. Generally, this study is interesting; however, there are still some concerns.

Comments:

  1. In "3.5. PEDV infection regulated autophagy-related gene mRNA", the authors analyzed the mRNA expression levels of autophagy-related genes ATG5, ATG12 and LAMP1. However, the change of mRNA expression level is usually not completely consistent with the change of protein expression level. The protein expression levels of ATG5, ATG12 and LAMP1 should be analyzed by western blot.

-> Thanks for kind comments. The activation and expression of both ATG5 and LAMP1 was confirmed by western blot analysis, and described in Figure 6 and 7. But, for ATG12, we confirmed mRNA level only, we added protein expression level using the anti-ATG12 antibody (Cell signaling, #4180) and confirmed by western blot. All the information described in materials and methods line 96, and added in Figure 6.

  1. Line 212-213, "LAMP1 mRNA expression at 4 hpi but upregulated it at 8 hpi. Interestingly, expression was significantly downregulated again at 12 hpi but significantly upregulated again at 24 hpi"; Line 224-226, "Additionally, we found that LAMP1 expression was downregulated both at 12 hpi and 24 hpi compared with those in mock-infected cells (Figure 6A-F)". Are these two results contradictory? It should be discussed in the discussion or results section.

-> Thanks for kind comments.  We added information in Discussion section (Line 281-288)

 We also confirmed that both genes were downregulated by PEDV infection at 4 hpi and 12 hpi, but significantly upregulated at 8 hpi and 24 hpi. Increase in mRNA doesn’t promising increase in protein expression, better confirmed protein expression by western blot. Therefore, we confirmed endogenous LAMP1 expression level by western blot by PEDV infection, and we found LAMP1 expression was reduced at both 12 hpi and 24 hpi as shown in Figure 5. The reason might be reduced protein expression by viral infection at 24 hpi recovered by the aspect of cell defense mechanism.

  1. Electron microscope is the gold standard for judging autophagosome or autolysosome. Electron microscope photos should be supplied in Figure 1, Figure 4 and Figure 7 if possible.

-> Thanks for kind comments. For electron microscope image, we think not essentially required this manuscript.

  1. Line 95, "mouse anti-PEDV" should be "mouse anti-PEDV antibody".

-> Thanks for kind comments. Following your advice, we changed line 95

  1. Line 143-144, "p value" the "p" should be italic.

-> Thanks for kind comments. Following your advice, we changed line 143-144.

  1. Figure 2B and 2C, the Mock and PEDV infection groups should be marked.

-> Thanks for kind comments Following your advice, we added legend beside to Figure 2B and 2C.

Round 2

Reviewer 2 Report

My concerns have been well answered.